# P2 Receptors Influence hMSCs Differentiation towards Endothelial Cell and Smooth Muscle Cell Lineages

**DOI:** 10.3390/ijms21176210

**Published:** 2020-08-27

**Authors:** Yu Zhang, Patrick Babczyk, Andreas Pansky, Matthias Ulrich Kassack, Edda Tobiasch

**Affiliations:** 1Department of Natural Sciences, Bonn-Rhein-Sieg University of Applied Sciences, D-53359 Rheinbach, Germany; Yu.Zhang@gmail.com (Y.Z.); Patrick.Babczyk@h-brs.de (P.B.); Andreas.Pansky@h-brs.de (A.P.); 2Institute of Pharmaceutical & Medicinal Chemistry, University of Dusseldorf, D-40225 Dusseldorf, Germany; matthias.kassack@hhu.de

**Keywords:** purinergic receptor, mesenchymal stem cell, endothelial cell differentiation, smooth muscle cell differentiation

## Abstract

Background: Human mesenchymal stem cells (hMSCs) have shown their multipotential including differentiating towards endothelial and smooth muscle cell lineages, which triggers a new interest for using hMSCs as a putative source for cardiovascular regenerative medicine. Our recent publication has shown for the first time that purinergic 2 receptors are key players during hMSC differentiation towards adipocytes and osteoblasts. Purinergic 2 receptors play an important role in cardiovascular function when they bind to extracellular nucleotides. In this study, the possible functional role of purinergic 2 receptors during MSC endothelial and smooth muscle differentiation was investigated. Methods and Results: Human MSCs were isolated from liposuction materials. Then, endothelial and smooth muscle-like cells were differentiated and characterized by specific markers via Reverse Transcriptase-PCR (RT-PCR), Western blot and immunochemical stainings. Interestingly, some purinergic 2 receptor subtypes were found to be differently regulated during these specific lineage commitments: P2Y4 and P2Y14 were involved in the early stage commitment while P2Y1 was the key player in controlling MSC differentiation towards either endothelial or smooth muscle cells. The administration of natural and artificial purinergic 2 receptor agonists and antagonists had a direct influence on these differentiations. Moreover, a feedback loop via exogenous extracellular nucleotides on these particular differentiations was shown by apyrase digest. Conclusions: Purinergic 2 receptors play a crucial role during the differentiation towards endothelial and smooth muscle cell lineages. Some highly selective and potent artificial purinergic 2 ligands can control hMSC differentiation, which might improve the use of adult stem cells in cardiovascular tissue engineering in the future.

## 1. Introduction

Cardiovascular diseases are responsible for the majority of deaths in industrialized countries. Currently, patients suffering from these diseases are often treated with bypass surgery which uses synthetic grafts to replace damaged vessels. Seeding patients’ own cardiovascular cells into synthetic or allogenic grafts can provide a non-thrombogenic, non-immunogenic, durable vessel substitute for bypass surgery. A suitable cell source for such a vascular conduit should conform to certain criteria. It should be autologous, available in large quantities, acquirable through minimally invasive techniques and have the capacity to differentiate and proliferate in a reproducible and controlled manner [1].

Human mesenchymal stem cells (hMSCs) fit all of these criteria and therefore have drawn attention as an effective cell source for seeding tissue-engineered grafts. hMSCs can be isolated from several different components of the body. These include more traditional sources such as bone marrow [2], fat [3], umbilical cord blood [4], skin [5,6], skeletal muscle [7] and newly characterized sources like dental follicle and heart tissues [8,9]. Liposuction material is regarded as an excellent hMSC source since it can be harvested easily from patients in large quantities and without inflicting significant pain [10]. These cells also have the capacity to differentiate along multiple cell lineages into numerous cell types. These include adipocytes, osteoblasts, chondrocytes [11] and recently endothelial and smooth muscle cells [9,10]. In order to ensure the most robust differentiation of hMSCs into the endothelial and smooth muscle cells needed for cardiovascular regenerative engineering, a more thorough understanding of the molecular mechanisms of hMSC differentiation into these cell types is essential. Proteins known as P2 receptors are thought to be important players in some of these mechanisms.

P2 receptors are known to participate in a variety of metabolic processes in different tissues and cells [12,13,14]. Importantly, they play a key role in many cardiovascular processes such as atherosclerosis, hypertension, angiogenesis and restenosis [15]. Fifteen P2 receptors have been characterized in human beings and are classified into seven ionotropic P2X subtypes (P2 × 1–7) and eight metabotropic P2Y subtypes (P2Y1, -2, -4, -6, -11, -12, -13 and -14) [16]. P2 receptors are activated by extracellular nucleotides and each subtype reacts differently and in a potency-order-dependent manner to its ligands [14]. In recent years, several publications have investigated the functional role of P2 receptors in stem cell behavior. For example, extracellular ATP, a prominent P2 ligand, is likely to regulate proliferation, migration and differentiation in embryonic stem cells [17], hematopoietic stem cells [18] and neuronal stem cells [19]. Accordingly, recent data indicate that P2 receptors are not only expressed in hMSCs but also directly influence hMSC differentiation towards adipocytes and osteoblasts [20,21]. However, the role of P2 receptors in stem cell endothelial and smooth muscle differentiation remains largely unknown. This study explores the role of P2 receptors in hMSC commitment towards endothelial and smooth muscle fates. We provide evidence that P2 receptors mediate hMSC endothelial and smooth muscle cell differentiation processes.

## 2. Results

### 2.1. Confirmation of the Human Mesenchymal Stem Cells (hMSC) Character

The isolated cells conformed to the minimal criteria for hMSCs as suggested by the International Society of Cellular Therapy [22]. The cells adhered to plastic plates and displayed a fibroblast-like morphology. They also expressed mesenchymal cell markers CD44, CD73, CD90 and CD105 and negative for CD14, CD45 by RT-PCR (Figure 1A) and flow cytometry analysis (Figure 1B and Appendix A). Surprisingly, these cells showed a partially positive expression for CD34. Isolated cells showed the potential to differentiate into mesodermal cell lineages: adipocytes and osteoblasts. Adipogenic differentiation was identified using Oil Red O staining of lipid deposits (Figure 1C) and osteogenic differentiation was verified by Alizarin Red S staining of calcium in the mineralized osteoblast matrix (Figure 1D). Additional characterizations were shown in the Appendix A. 

### 2.2. Differentiation of hMSCs towards Endothelial and Smooth Muscle Cells

Differentiated endothelial-like cells displayed increased gene and protein expression of endothelial cell specific markers eNOS, Flk-1 and PECAM-1 when compared with undifferentiated cultures as ascertained through RT-PCR (Figure 2A) and Western Blot (Figure 2B). Differentiated cells also showed positive immunochemical staining for eNOS (Figure 2D). Characteristic endothelial activity was documented in differentiated endothelial-like cultures by DiI-ac-LDL uptake (Figure 2C) and tube formation (Figure 2E) experiments. These findings indicate successful differentiation into endothelial cells. In differentiated smooth muscle-like cells, increased expression of smooth muscle cell specific markers α-SMA, SM22α and calponin was detected by RT-PCR (Figure 2F) and later confirmed with Western blot (Figure 2G) and immunochemical staining with calponin (Figure 2H), indicating successful smooth-muscle differentiation.

### 2.3. P2 Receptor Expression during Endothelial and Smooth Muscle Cell Differentiation

RT-PCR revealed that P2X5, P2Y1, P2Y2, P2Y4, P2Y11 and P2Y14 were up-regulated in endothelial-differentiated cells when compared with undifferentiated cells while P2Y6 was down-regulated (Figure 3A). Western blot and flow cytometry showed a similar results regarding protein expression (Figure 3B–D, Figure 4 and Appendix A). During smooth muscle differentiation, P2X1 and P2Y14 gene expression increased and P2X3, P2X7 and P2Y1 expression decreased (Figure 3C). Protein expression followed the same trend (Figure 3D–F, Figure 4 and Appendix A).

### 2.4. The Influence of Natural and Artificial P2 Agonists and Antagonists on Endothelial and Smooth Muscle Cell Differentiation

hMSCs were differentiated in the presence of the naturally-occurring P2 receptor agonists ATP, UTP, ADP, UDP, UDP-glucose and universal antagonists suramin and PPADS to investigate these molecules’ influence on endothelial and smooth muscle cell differentiation. The highly potent and selective P2 receptor agonist MRS2365 (P2Y1), as well as antagonists MRS2500 (P2Y1), RO-3 (P2X3), A-740003 (P2X7) were used to investigate specific P2 subunit function in the differentiation process. ATP, ADP and UDP-glucose significantly increased tube formation of endothelial-differentiated hMSCs in matrigel matrices (Figure 5A) and increased eNOS positive cells (Figure 5F). Administration of ATP and ADP resulted in far fewer calponin positive cells when compared with administration of other agonists, while co-administration of ATP or ADP with suramin and PPADS increased calponin-positive cell counts (Figure 5B). Further, A-740003 and MRS 2500 enhanced calponin expression while RO-3 seemed to have no effect (Figure 5C–F, Appendix A).

In order to evaluate the effect of endogenous nucleotides, apyrase enzyme was added into endothelial and smooth-muscle differentiation media. Apyrase is a diphosphohydrolase which can hydrolyze extracellular nucleotides, in effect depriving cells of signals carried by cell-released nucleotides. hMSC-derived endothelial cells differentiated in media containing 5U/mL apyrase displayed a decrease in tube formation, indicating a less-robust differentiation (Figure 5D). Yet, when it was used to differentiate hMSC-derived smooth muscle cells, it enhanced the percentage of calponin positive cells post-differentiation (Figure 5E). These findings suggest that the release of endogenous nucleotides from hMSC during differentiation plays a pivotal role in the regulation of endothelial and smooth muscle cell differentiation.

### 2.5. The Possible P2 Underlying Signaling Pathway Involved in Endothelial Cell and Smooth Muscle Cell Differentiation

A protein kinase activity assay was used to study the possible underlying P2 signaling pathways. During endothelial cell differentiation, the pathways ERK1/2, β-catenin, STAT1, RSK1/2 and c-jun were inhibited (Figure 6A, Appendix A) while TOR and HSP27 were activated. After smooth muscle cell differentiation (Figure 6B, Appendix A), AKT, p53, paxillin and c-jun were activated while AMPKα1 was inhibited. The decrease of ERK1/2 phosphorylation (Figure 6C) in endothelial differentiation and the increase of AKT phosphorylation (Figure 6D) relative to undifferentiated cultures was further confirmed with Western blot. p38 phosphorylation but not p38α was also enhanced by endothelial differentiation.

## 3. Discussion

To produce clinically useful artificial constructs to replace cardiovascular structures such as blood vessels, it is critical to identify an appropriate cell source which is autologous and can differentiate into the necessary cell types. Recent studies show the potential of MSCs derived from different sources to trans-differentiate (the differentiation from a cell type of one germ-layer to a cell type of the other germ-layer) into endothelial cells [12,23] and differentiate into smooth muscle cells [24]. Trans-differentiation might occur because MSC populations are not composed of a single type of cell, but a diverse mixture of different lineage progenitors [25]. In addition to their broad capacity for differentiation, MSCs have also been shown to invoke a less violent immune response than other types of stem cells such as ESCs and iPSCs in vivo [26]. Moreover, MSCs are less likely to form teratomas, and are more likely to secrete beneficial cytokines and show higher migration after implantation [27].

The MSCs derived from adipose tissue constitute a more attractive cell source than MSCs derived from other regions. MSCs sourced from liposuction material are easily accessible, and liposuction-material derived MSCs can be harvested in abundance with a less invasive procedure than is required to acquire MSCs from other sources [28]. They also have a remarkable stability with respect to the age of their source, with cells acquired from elderly patients being equally abundant and harboring the same capacity for differentiation as those acquired from younger donors [23]. Comparative analysis of MSCs derived from bone marrow, adipose and dermal tissue showed that adipose tissue-derived MSCs secreted the greatest amount of beneficial cytokines such as VEGF-D and enhanced HMEC tube formation to the greatest extent. Such findings indicate that these cells had stronger ability to repair the local damaged tissue after injecting in vivo than other MSC types [29]. Taken together, this evidence indicates that hMSCs derived from adipose tissue are a reliable and promising cell source for use in cardiovascular tissue engineering.

In this study, P2Y4 and P2Y14 were up-regulated after differentiation into endothelial and smooth muscle cells, which are cell types of vascular lineage. Another study found these receptor subtypes to be down-regulated after differentiation into adipocytes and osteoblasts, cell types of non-vascular lineage [30]. Lineage commitment occurs early in the differentiation process. Accordingly, features conserved across different cell types belonging to same lineage are likely to arise earlier. Therefore, these findings suggest that these two receptors may be important in the early commitment for hMSC differentiation towards different cell lineages. P2Y1, P2Y2 and P2Y11 were also found to be up-regulated after endothelial differentiation. ATP and ADP significantly increased tube formation in matrigel scaffolds while UTP had no effect, suggesting that P2Y1 and P2Y2 might be particularly important for endothelial differentiation. P2Y6 was the only P2 subtype that was downregulated after endothelial differentiation. Xiao and colleagues demonstrated that endothelial progenitors are negative for P2Y6, which suggests that P2Y6 may be an important regulator of cellular identity in mature endothelial progenitors, and could be used to efficiently characterize endothelial progenitors. They also found that P2Y2 receptors were more likely to be activated by ATP than by UTP [31]. 

P2Y1 was down-regulated after smooth muscle differentiation while it was up-regulated after endothelial differentiation. P2Y1 selective agonist MRS 2365 inhibited calponin expression while its antagonist MRS2500 enhanced its expression. This indicates that P2Y1 may be a key factor in directing MSC differentiation towards endothelial and smooth muscle cells. Both P2X3 and P2X7 were down-regulated in smooth muscle differentiation. The P2X3 artificial antagonist RO-3 had no significant effect in the differentiation, while P2X7 antagonist A-740003 lead to distinct increases in calponin expression over undifferentiated cultures, indicating a differentiation-enhancing effect. The reason for P2X7′s potency might be that P2X7 has an intracellular C-terminal tail containing several binding domains which can initiate a variety of signaling pathways such as ERK1/2, Rho and JNK [32]. It is known that proliferation and differentiation oppose each other in stem cells, and thus P2X7 stimulation might enhance the differentiation by inhibiting proliferation [10]. 

The application of the nucleotide hydrolyzing enzyme apyrase decreased tube formation while enhancing calponin expression, indicating that endogenous nucleotide release was heavily involved in both endothelial and smooth muscle cell differentiation. Other studies have documented the extensive effects of endogenous nucleotides on hMSC signaling and differentiation. Our recent data also reported that cellular release of nucleotides can enhance hMSC osteogenic differentiation but inhibit adipogenic differentiation [20]. hMSCs have been shown to release ATP, which induces intracellular calcium oscillations in an autocrine/paracrine fashion [33]. Another study showed that mature osteoblasts release as much as seven times the ATP released by immature osteoblasts, suggesting that stem cells might release ATP in a differentiation-dependent manner [34]. Taken together, these data could suggest that release of endogenous nucleotides might directly influence differentiation through a positive feedback loop, as increases in extracellular nucleotides promote differentiation, which in turn spurs the release of more nucleotides. P2Y receptors are able to trigger many different signaling pathways due to their abundance of different subtypes. For example, P2Y1 is coupled with G-protein Gq and can activate PI3K/AKT/NF-κB or MKK/p38/CREB pathways, while P2Y11 is the only Gs-coupled P2Y and can initiate the cAMP/ERK/c-jun pathway. P2Y11 has the unique feature of being coupled with both Gq and Gs; similarly, P2Y4 is coupled with Gq and Gi. Blocking the ERK/MAPK pathway can improve bone marrow derived hMSC smooth muscle differentiation [35]; however, such blockage has the opposite effect of inhibiting endothelial differentiation in bone marrow derived progenitor cells [36]. Differentiation dramatically altered the activity of several pathways, as assessed through kinase phosphorylation. The ERK1/2 pathway showed reduced activation in endothelial differentiation, while p38 was more active. AKT was up-regulated after smooth muscle differentiation. It can be hypothesized that signals that maintain stem-cell potency and block differentiation are replaced by those that block anti-differentiation signals and promote differentiation through the interaction of extracellular nucleotides with P2 receptors. This hypothesis requires further investigation.

Different P2 subtypes are not uniformly expressed in vascular cells: P2Y1, P2Y2, P2Y4, P2Y11 are prominently expressed in endothelial cells and P2Y1, P2Y2 and P2Y12 predominated in smooth muscle cells [37]. Their roles in various vascular pathologies such as atherosclerosis, hypertension and vascular pain have been demonstrated [38]. Several P2 artificial ligands are already used as clinical drugs. For example, Plavix, which is an antagonist for P2Y12, is the top-selling drug for hypertension. However, how to utilize these P2 artificial ligands to improve the quality of engineered vascular grafts is nearly unexplored [22]. One of the most important requirements for creating re-cellularized grafts is the robust and comprehensive control of stem cell differentiation towards desired cell lineages [39]. By profiling the differentiation-relevant P2 expression patterns and stimuli in adipose tissue-derived hMSCs, our data give insight into how to use artificial P2 ligands to improve the quality of re-cellularized scaffolds. There have been several approaches to differentiate hMSCs towards the vascular lineages, e.g., the usage of vectors [40], sphingosine 1-phosphate [41] or the conduction of different coculture models [42,43,44]. All these strategies reached the state of endothelial progenitor cells (EPCs) or used EPCs as cell source. Recent strategies are based on the variation of the inducing medium to trigger the differentiation [45].

In summary, our results clearly demonstrate that P2 receptors directly influence both endothelial and smooth muscle cell differentiation. Our data suggest that P2 receptors play a crucial role in systematically controlling stem cell fate in different stages (Figure 7). In light of previous data on osteogeneic and adipogenic MSC differentiation, our findings indicate that P2Y4 and P2Y14 are involved in the early lineage commitment (pre-adipocytes, -osteobalsts). Additionally, they suggest that P2Y1 seems to be the key factor in determining MSC differentiation towards either (pre-) endothelial or (pre-) smooth muscle cells. As a result of the importance of P2 receptors in controlling hMSC differentiation, extracellular nucleotides and P2 artificial ligands might be prominent candidates to control stem cell differentiation processes and ultimately could greatly improve cardiovascular tissue engineering in the future.

## 4. Materials and Methods 

### 4.1. Cell Isolation, Culture and Characterization

Human liposuction aspirates were obtained from several healthy female donors undergoing plastic surgery. The age group 20–44 was selected in order to cover a representative range of ages. The isolation and characterization procedures were performed according to our previous study [21]. 

In short, cells were isolated by collagenase digestion and different washing steps. After the isolation the gained MSC were cultured in DMEM supplemented with stable L-Glutamine and 10% FBS. Characterization of the cells was performed using RT-PCR and flow cytometry, checking for positive and negative MSC marker according to the minimal criteria of the International Society for Cellular Therapy [22].

### 4.2. Endothelial and Smooth Muscle Cell Differentiation

All differentiation experiments were performed between cell passage numbers 2–3. To induce endothelial cell differentiation, the medium was replaced with endothelial cell growth medium 2 (EGM-2) (PromoCell GmbH, Heidelberg, Germany) when hMSCs reached 70% confluence. This medium contained 2% FCS, vascular endothelial cell growth factor (VEGF), basic fibroblast growth factor (bFGF), insulin-like growth factor (IGF) and epidermal growth factor (EGF). The medium was changed every 2 days. The whole induction lasted 2 weeks. 

For the smooth muscle cell differentiation, hMSCs were cultured with smooth muscle inducing medium (SMIM) when cells reached 80% confluence. This medium consisted of DMEM medium supplemented with 2% FCS, 10 ng/mL platelet derived growth factor (PDGF-ββ) (PromoCell), 5 ng/mL Transforming growth factor-β1 (TGF-β1) (PromoCell) and 30 μg/mL heparin (Biochrom AG, Berlin, Germany). The induction process lasted 2 weeks with medium exchange occurring every 2 days. 

### 4.3. Semiquantitative RT-PCR

RNA extraction was conducted with innuSOLV RNA reagent (Analytic Jena, Jena, Germany) after differentiation. cDNAs were synthesized from 1–2 µg total RNA. Primers (Appendix A) for endothelial and smooth muscle cell specific markers (endothelial nitric oxide synthase (eNOS), platelet endothelial cell adhesion molecule 1 (PECAM-1), vascular endothelial growth factor receptor 2 (Flk-1), calponin, transgelin (SM22α) and alpha-smooth muscle actin (α-SMA)) were designed to exclude cross-reactions as shown in Appendix A. The cell line HMEC-1 was used as a positive control for endothelial cell differentiation and bovine aortic smooth muscle cells were used as a positive control for smooth muscle cell differentiation. P2 receptor primers and positive controls were the same as those used in our previous study [21]. In short, RT-PCR for bActin was done to balance all samples (23 cycles) and then these have been used for markers and receptors (45 cycles).

### 4.4. Western Blot

The process was carried out in accordance with our previous study [21]. The anti-eNOS, PECAM-1, Flk-1, calponin, smooth muscle actin, transgelin-2, P2X1, P2X3, P2X5, P2X7, P2Y1, P2Y2, P2Y4, P2Y6, P2Y11 and P2Y14 antibodies were purchased from Santa Cruz Biotechnology. Protein extracts from the cell lines HMEC-1, C2 and MG-63 cells were applied as positive controls for the Western blot. Protein density was analyzed by Image J software (ImageJ 1.50i, Wayne Rasband National Institutes of Health, Bethesda, .MD USA) for P2 expression after differentiation. 

### 4.5. Immunochemical Staining and Quantification

Briefly, differentiated cells were fixed with 4% formaldehyde at room temperature for 15 min. Cells were washed with PBS. The cells were incubated with first-antibody (Santa Cruz, Heidelberg, Germany) at 4 °C overnight. After being washed with PBS, fixed cultures were incubated with a secondary-antibody (Alexa Fluor-488, ThermoFisher Scientific, Schwerte, Germany), at room temperature for 1 h. DAPI counter stain followed for 5 min incubation and after this, cultures were rinsed again with PBS and fluorescence measurements were performed using an immunofluorescence microscope (Zeiss Observer D.1, Jena, Germany) using same illumination time. Quantification was done using Image J software.

### 4.6. Flow Cytometry Analysis

Mesenchymal stem cells were grown to 90% confluency, harvested and fixed in 4% formaldehyde solution for 10 min at 37 °C. After 1 min of cooling on ice, cells were centrifuged (5 min, 200× *g*) and resuspended in incubation buffer (5% BSA in PBS). Staining of 5000 cells/vial with fluorochrome-labeled antibodies for positive markers CD44, CD90, CD105 and negative markers CD14, CD34 and CD45 (Beckmann &Coulter, dilution according to manufacturer’s instructions) was conducted for 20 min at room temperature followed by 3 washing steps in incubation buffer. After centrifugation (5 min, 200× *g*) cells were re-suspended in PBS for flow cytometry analysis. Analysis was performed with a Cytomics FC500 flow cytometer (Beckmann & Coulter, Krefeld, Germany) and software. 

To investigate the expression profile of purinergic receptors an indirect flow cytometry analysis was performed. Therefor mesenchymal stem cells from human adipose tissue were differentiated towards endothelial cells and smooth muscle cells for 7 days to investigate the expression profile of P2Y1, P2Y4 and P2Y4 and for 14 days to investigate the expression profile of P2X5, P2X7, P2Y6 and P2Y11. Cells were fixed for 10 min at 37 °C with 4% formaldehyde and 1 min cooling on ice. Permeabilization of the cells was conducted in 90% Methanol for 10 min at −20 °C. Staining of 5000 cells was performed by using primary antibodies (P2-Receptor antibodies; Alomone Labs, MyH-11; Santa Cruz; dilution according to manufacturer’s instructions) and fluorochrome-labeled secondary antibodies (Alexa Fluor488; dilution according to manufacturer’s instructions) for 1 h incubation of each sample at room temperature separated by washing 3 times with incubation buffer. Cells were re-suspended in PBS for flow cytometry analysis.

### 4.7. Determination of DiI-Ac-LDL Uptake

MSCs were cultured in DMEM and EGM-2 in a 48-well plate with the same cell density as described in Section 2.2 for 2 weeks. Cells were incubated with 10 µg/mL DiI-labeled acetylated-low density lipoprotein (DiI-Ac-LDL; Sigma-Aldrich, Taufkirchen, Germany) at 37 °C for 4 h, and directly viewed with an immunofluorescence microscope (Zeiss, Ulm, Germany).

### 4.8. Matrigel Tube Formation Assay

The 96-well plate wells were coated with 100 µL per well of 10 mg/mL Growth Factor Reduced Matrigel (GFR-Matrigel, BD Biosciences, Heidelberg, Germany) and incubated for 1 h at 37 °C to solidify. MSCs were seeded onto the GFR-Matrigel at a density of 5 × 10^4^/cm^2^. After 3 days incubation, the formation of tube-like structures was examined with a phase-contrast microscope. 

### 4.9. Natural and Artificial Agonist and Antagonist Stimulation

hMSCs were cultured in DMEM, EGM-2 and SMIM medium supplemented with natural nucleotides (ATP, UTP, ADP, UDP-glucose) and universal P2 receptor antagonists (suramin and PPADS). A concentration of 100 μM nucleotides was used to antagonize P2-receptors specifically as accomplished in our previous study [18], and no apoptosis signals were detected during the differentiation process. Twice each week, half of the medium was replaced by fresh medium containing a 200 μM concentration of agonists or antagonists. 

Selective artificial agonists and antagonists have not been determined for all P2 receptors investigated in this study [23]. The highly selective and potent P2 × 3 antagonist RO-3 (pIC_50_ = 7.0, Tocris), P2 × 7 antagonist A-740003 (IC_50_ = 40 nM, Sigma), P2Y1 agonist MRS2365 (EC_50_ = 0.4 nM, Tocris) and P2Y1 antagonist MRS2500 (Ki = 0.78 nM, Tocris) were used for specifically activating or inhibiting target P2 receptors. Tube formation experiments and calponin staining were performed as described in Section 4.5 and Section 4.7 to evaluate the effects of P2 agonist and antagonist stimulation.

### 4.10. Nucleotide Cleavage

A total of 5 U/mL apyrase (Sigma) was administrated into the DMEM, EGM-2 and SMIM medium during the differentiation process. Half of the medium was replaced by fresh medium supplemented with twice this concentration of apyrase two times a week. Tube formation experiments and calponin staining were performed as described in Section 4.5 and Section 4.7. 

### 4.11. Proteome Profiler Antibody Assay

Human phosphor-kinase array kits were purchased from R&D System (Minneapolis, MN, USA). Undifferentiated cells and differentiated cells undergoing endothelial and smooth muscle differentiation were analyzed separately. Relative levels of phosphorylation of 46 kinase phosphorylation sites were performed and the density was compared by using Image J software. 

### 4.12. Statistical Analysis

The data are shown as mean ± SD. The experiments were performed in triplicate for at least three donors. Differences between groups were assessed using the Student’s *t* test by SPSS software. The following thresholds were considered to be significant: (* *p* < 0.05), (** *p* < 0.01) or (*** *p* < 0.001).

### 4.13. Ethics Sstatement

The use of human liposuction materials for isolation, differentiation and characterization of human mesenchymal stem cells was approved by the Friedrich-Wilhelms University of Bonn, medical faculty, ethics committee (209/04, 05.11.2004). All subjects gave written informed consent.

## Figures and Tables

**Figure 1 ijms-21-06210-f001:**
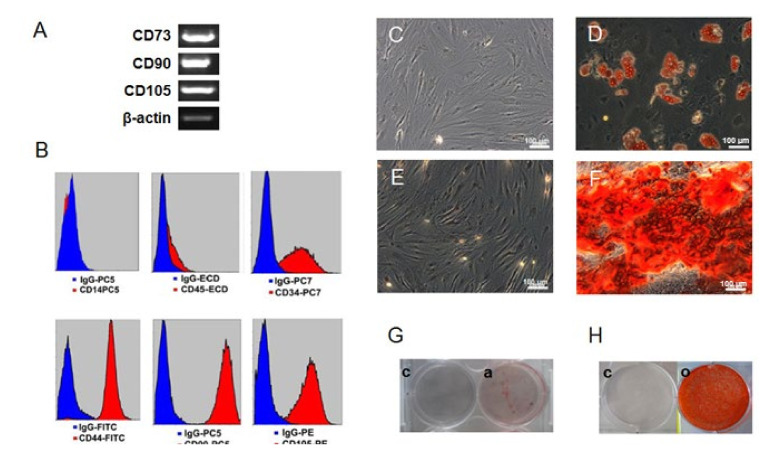
Verification of mesenchymal stem cell character. Isolated cells were positive for CD44, CD73, CD90 and CD105 and negative for CD14, CD45 by RT-PCR (**A**) and flow cytometry (**B**). Interestingly, these cells partially expressed CD34. Adipogenic differentiation was evaluated by Oil Red O staining of undifferentiated (**C**) and adipogenic differentiated cells (**D**). The difference was also visible from petri-dish (**G**). Mineralization during osteogenic differentiation was performed by Alizarin Red S staining of un-differentiated (**E**) and osteogenic differentiated cells (**F**). The difference was also compared between petri dish grown cultures (**H**). Human mesenchymal stem cells (hMSC) markers and specific stainings were chosen in accordance with the International Society for Cellular Therapy [19]. β-actin served as internal control. Each picture represents one donor out of 3; each yielded similar results. c, un-differentiated cells; a, adipogenic differentiated cells; o, osteogenic differentiated cells.

**Figure 2 ijms-21-06210-f002:**
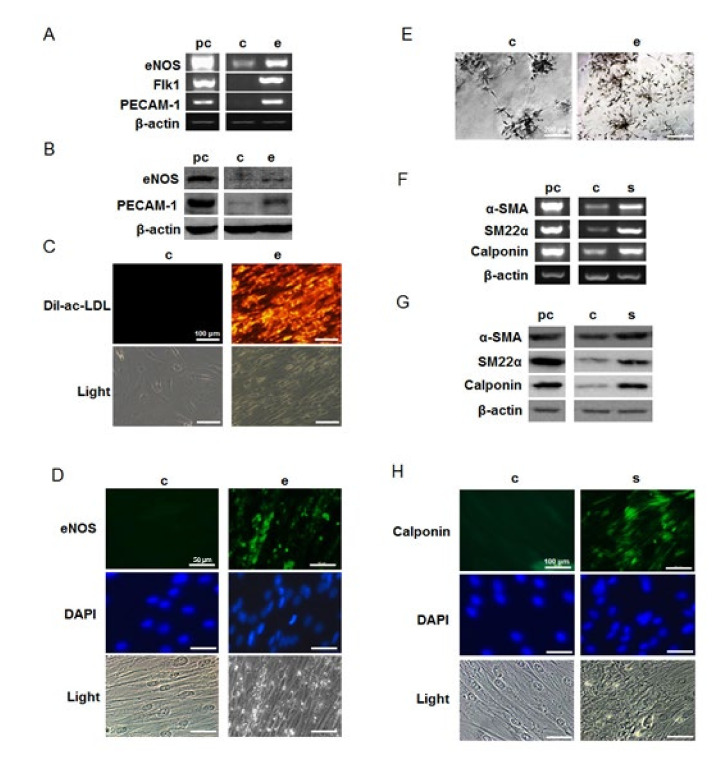
Induction of endothelial and smooth muscle cell markers when hMSCs were grown in endothelial and smooth muscle cell differentiation media. Endothelial differentiated cells showed up-regulation of endothelial cell markers eNOS, flk-1 and PECAM-1 by RT-PCR (**A**) and Western blot (**B**); immunochemical analysis showed differentiated cells positively expressed eNOS (**D**). Differentiated cells could uptake DiI-ac-LDL (**C**) and form more tubes in Matrigel better than un-differentiated cells (**E**); up-regulations of smooth muscle cell specific marker α-SMA, SM22α and calponin were found in smooth muscle differentiated cells in both gene (**F**) and protein (**G**) levels. Differentiated cells showed positive immunofluorescent staining for calponin (**H**); pc, positive control; c, un-differentiated cells; e, endothelial differentiated cells; s, smooth muscle differentiated cells.

**Figure 3 ijms-21-06210-f003:**
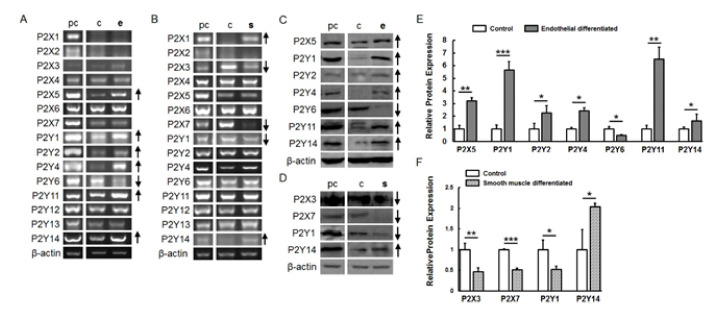
Regulation of P2 receptor expression after endothelial and smooth muscle cell differentiation. P2 receptor gene expression after endothelial (**A**) and smooth muscle differentiation (**B**). P2 receptor protein expression after endothelial (**C**) and smooth muscle differentiation (**D**) by Western blot with β-actin as the internal control. Graphs (**E**,**F**) represent the quantification of bands normalized with β-actin in undifferentiated and differentiated cells. pc, positive control; c, un-differentiated cells; e, endothelial differentiated cells; s, smooth muscle differentiated cells. ↑ indicates an up-regulation of the P2 receptor subtype during differentiation, and ↓ indicates a down-regulation. The data shown represent 1 donor out of 3). (* *p* < 0.05; ** *p* < 0.01; *** *p* < 0.001).

**Figure 4 ijms-21-06210-f004:**
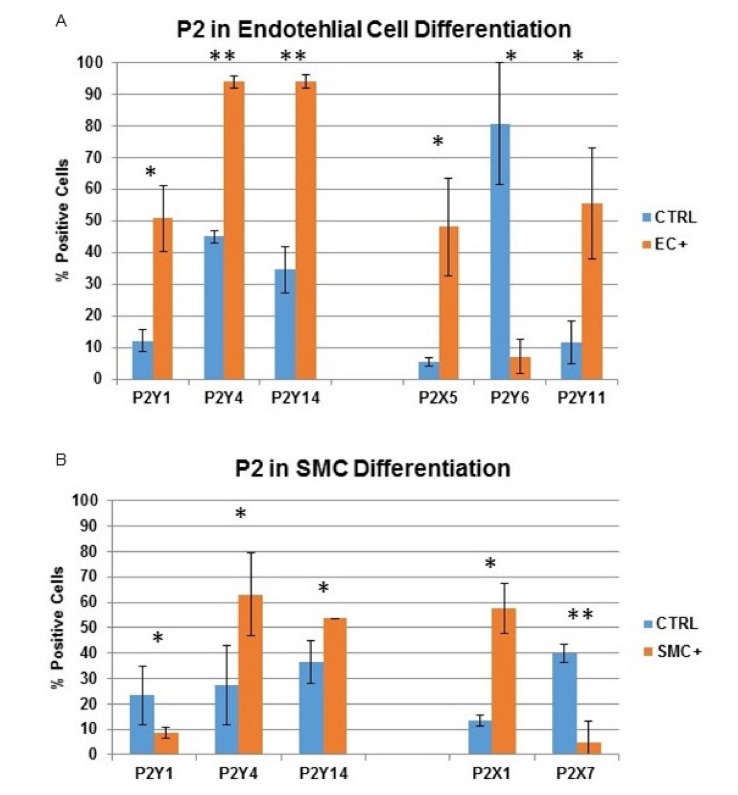
Indirect flow cytometry analysis of P2 receptor expression after endothelial and smooth muscle cell differentiation. (**A**) P2 receptor expression of undifferentiated and endothelial cell differentiated MSC after 7 days (P2Y1, P2Y4 and P2Y14) and after 14 days (P2X5, P2Y6 and P2Y11); (**B**) P2 receptor expression of undifferentiated and smooth muscle cell differentiated MSC after 7 days (P2Y1, P2Y4 and P2Y14) and after 14 days (P2X1 and P2X7). (* *p* < 0.05; ** *p* < 0.01).

**Figure 5 ijms-21-06210-f005:**
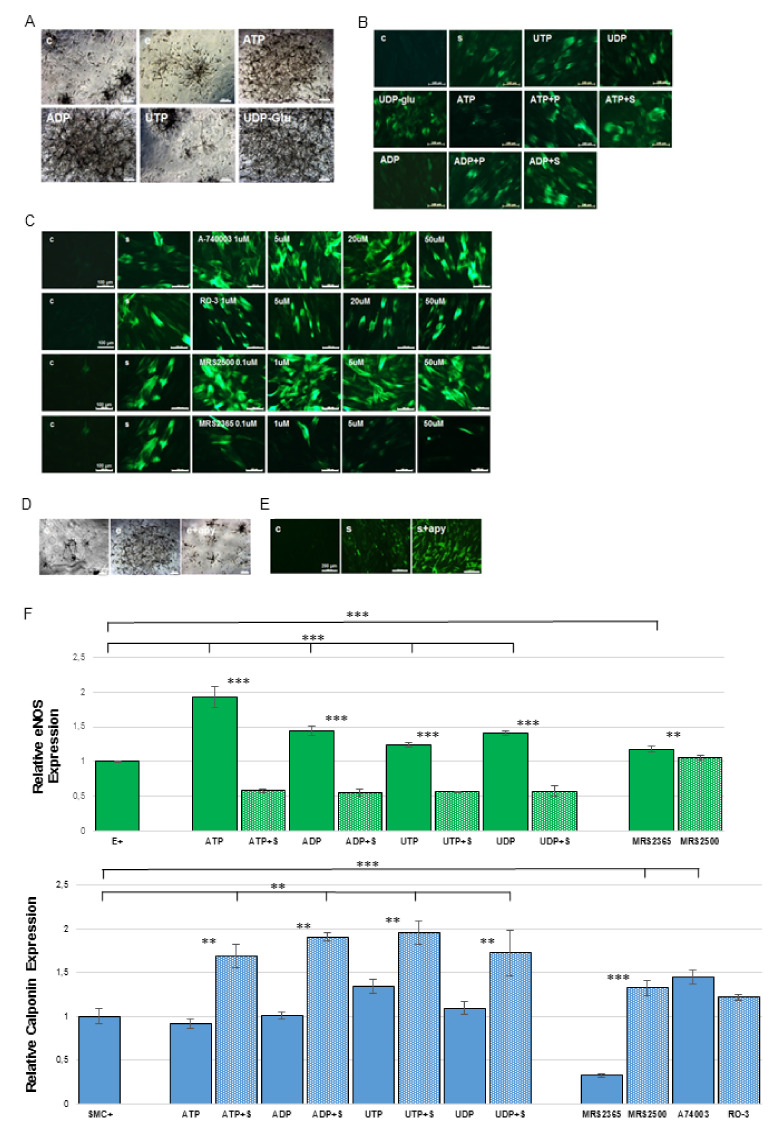
Natural and artificial P2 receptor agonists and antagonists influenced hMSC differentiation towards endothelial cells and smooth muscle cells. Endothelial and smooth muscle cell differentiated cells were evaluated by tube formation assay, eNOS and calponin staining separately. (**A**) Both ATP and UDP-glucose can significantly enhance tube formation. Scale bar 200 µm; (**B**) ATP and ADP decreased calponin expression while PPADS and suramin reversed these effects. Scale bar 100 µm; (**C**,**F**) P2X7 artificial antagonist A74003 and P2Y1 antagonist MRS2500 enhanced the number of calponin-positive cells, P2Y1 agonist MRS2365 nearly inhibited smooth muscle cell differentiation, while P2X3 antagonist RO-3 seemed to have no effect. Scale bar 100 µm; (**D**) Endogenous nucleotide released during the endothelial cell and smooth muscle cell differentiation increased tube formation Scale bar 200 µm; (**E**) while inhibiting smooth muscle cell differentiation. Scale bar 200 µm; (**F**) Quantification of endothelial and smooth muscle cell differentiation and treatment with natural and artificial agonist and antagonists. ** *p* < 0.05; *** *p* < 0.001. (Immunofluorescent pictures see Appendix A).

**Figure 6 ijms-21-06210-f006:**
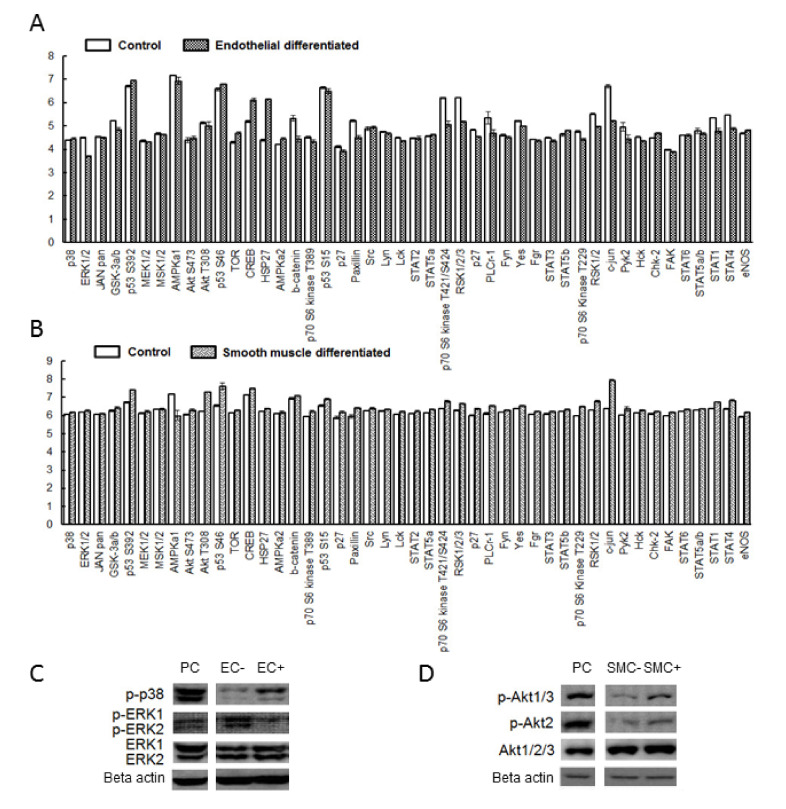
The possible underlying signaling transduction pathways in endothelial and smooth muscle cell differentiation. Human phosphor-kinase array was utilized to evaluate 46 kinase phosphorylation sites after endothelial (**A**) and smooth muscle (**B**) differentiation. ERK1/2, β-catenin, STAT1, RSK1/2 and c-jun were inhibited while TOR and HSP27 were activated after endothelial differentiation (A). p53, Akt, paxillin and c-jun were activated while AMPKα1 was inhibited after smooth muscle differentiation (B). In addition, Western blot analysis confirmed that p38 was activated while ERK1/2 was inhibited in endothelial differentiation (**C**); Akt was activated in smooth muscle cell differentiation (**D**).

**Figure 7 ijms-21-06210-f007:**
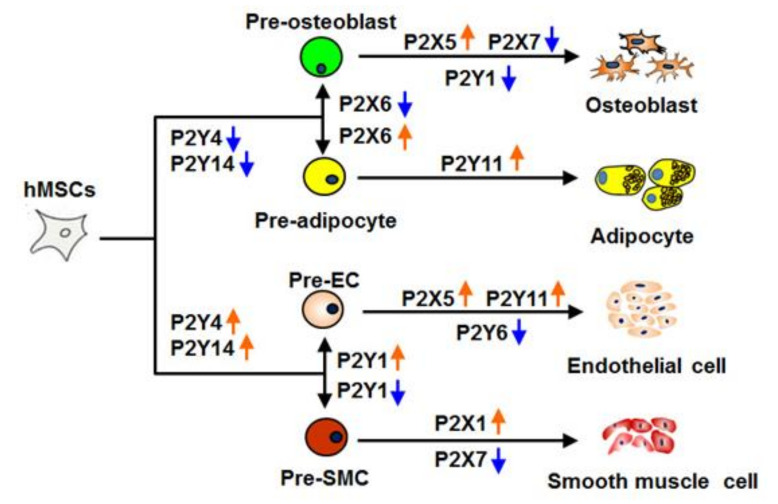
Model of P2 receptor function in MSC commitment. P2Y4 and P2Y14 are indicated to involve in the early differentiation stages, whereas P2Y1 might be a key factor in differentiation towards endothelial and smooth muscle cells. P2X5, P2Y4 andP2Y6 appear to be important in endothelial differentiation, while P2X7 appears to be involved in smooth muscle differentiation.

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
