# Peer review of "P2 Receptors Influence hMSCs Differentiation towards Endothelial Cell and Smooth Muscle Cell Lineages"

_ijms, 2020, doi:10.3390/ijms21176210_

Round 1

Reviewer 1 Report

The authors have examined the role of P2 receptors in hMSC commitment towards endothelial and smooth muscle fates and provided evidence that P2 receptors mediate hMSC endothelial and smooth muscle cell differentiation processes. P2Y4 and P2Y14 are indicated to involve in the early differentiation stages, whereas P2Y1 might be a key factor in differentiation towards endothelial and smooth muscle cells. The authors have concluded that extracellular nucleotides and P2 artificial ligands might be prominent candidates to control stem cell differentiation processes and ultimately could greatly improve cardiovascular tissue engineering in the future.

Abstract: well written

Introduction: well written

Results: well written

Discussion: well written

Figure6C and D:  beta actin is needed.

Reviewer 2 Report

This is a well written and well conducted manuscript regarding the possible role of purinergic P2 receptors in differentiation of human mesenchimal stem cells towards endothelial and smooth muscle cells.

Not many papers are present in literature regarding this topic. Therefore this manuscript shows new interesting data

Data presented in this manuscript are quite convincing and key roles for some of these receptors in the differentiation process are suggested

Some few modifications are required

Figure 6 should be presented better since some details are lost; in addition, endothelial differentiated cells also showed increase of p70 S6 kinase T421/S464, a detail that is missed in the description

A minor spell check is required, e.g. 

line 82 "osteoblasts" instead of "osteoblast"

line 94: "petri dish grown cultures" instead of "petri dish grown cultures petri dish"

line 112: "cultures as ascertained through RT-PCR (Figure 2A)" instead of "cultures (Figure 2A) as ascertained through RT-PCR "

line 164: "Endogenous" instead of Endegenous"
